# Monoubiquitination by the human Fanconi anemia core complex clamps FANCI: FANCD2 on DNA in filamentous arrays

Winnie Tan[1,2], Sylvie van Twest[1], Andrew Leis[3], Rohan Bythell-Douglas[1], Vincent J Murphy[1], Michael Sharp[1], Michael W Parker[3,4], Wayne Crismani[1,2], Andrew J Deans[1,2]*

[1]Genome Stability Unit, St. Vincent's Institute of Medical Research, Fitzroy, Australia; [2]Department of Medicine (St. Vincent's Health), The University of Melbourne, Melbourne, Australia; [3]Bio21 Institute, University of Melbourne, Parkville, Australia; [4]Structural Biology Unit, St. Vincent's Institute of Medical Research, Fitzroy, Australia

**Abstract** FANCI:FANCD2 monoubiquitination is a critical event for replication fork stabilization by the Fanconi anemia (FA) DNA repair pathway. It has been proposed that at stalled replication forks, monoubiquitinated-FANCD2 serves to recruit DNA repair proteins that contain ubiquitin-binding motifs. Here, we have reconstituted the FA pathway in vitro to study functional consequences of FANCI:FANCD2 monoubiquitination. We report that monoubiquitination does not promote any specific exogenous protein:protein interactions, but instead stabilizes FANCI: FANCD2 heterodimers on dsDNA. This clamping requires monoubiquitination of only the FANCD2 subunit. We further show using electron microscopy that purified monoubiquitinated FANCI: FANCD2 forms filament-like arrays on long dsDNA. Our results reveal how monoubiquitinated FANCI:FANCD2, defective in many cancer types and all cases of FA, is activated upon DNA binding.

**\*For correspondence:**
adeans@svi.edu.au

**Competing interests:** The authors declare that no competing interests exist.

## Introduction

Fanconi anemia (FA) is a devastating childhood syndrome that results in bone marrow failure, leukemia and head and neck cancers (*Gillio et al., 1997*; *Butturini et al., 1994*). FA is caused by inheritance of one of 22 dysfunctional FA genes (FANCA-FANCW) (*Tan and Deans, 2017*). Absence of any one member of the pathway causes genome instability during DNA replication, which results in mutagenic (cancer-causing) DNA damage and hypersensitivity to chemotherapeutic (normal and cancer-killing) DNA damage (*Deans and West, 2011*). Central to the FA pathway is the conjugation of ubiquitin to FANCI:FANCD2 (ID2) complexes (*Walden and Deans, 2014*). ID2 monoubiquitination is critical to prevention of bone marrow failure in FA, but it is currently unknown how ID2-ub differs in its function to ID2. Several proteins have been proposed to specifically bind FANCI[Ub] or FANCD2[Ub] but not the un-ubiquitinated proteins (*Smogorzewska et al., 2010*; *Lachaud et al., 2014*). For example, FAN1 nuclease was proposed to interact with FANCD2[Ub] via its ubiquitin-binding domain (UBZ) (*Smogorzewska et al., 2010*), whereas recruitment of SLX4 endonuclease to the interstrand crosslink site was shown to be dependent on FANCD2 ubiquitination (*Klein Douwel et al., 2014*). However, support for these interactions is limited to analysis of ubiquitination-deficient (K > R) mutants, rather than evidence for direct ubiquitin-mediated protein interactions.

The retention of FANCD2 in chromatin foci is dependent on its monoubiquitination by a 'core complex' of Fanconi anemia proteins (*Walden and Deans, 2014*). FANCI and the FA core complex are required to generate FANCD2-foci that mark the location of double strand breaks, stalled

**eLife digest** Bone marrow is the spongy tissue inside bones that produces blood cells. Fanconi anemia is the most common form of inherited bone marrow death and affects children and young adults. In this disease, bone marrow cells cannot attach a protein tag called ubiquitin to another protein called FANCD2. When DNA becomes damaged, FANCD2 helps cells to respond and repair the damage but without ubiquitin it cannot do this correctly. Without ubiquitin linked to FANCD2 bone marrow cells die from damaged DNA. Another protein, called FANCI, works in partnership with FANCD2 and also gets linked to ubiquitin.

Tan et al. studied purified proteins in the laboratory to understand how linking ubiquitin changes the behavior of FANCD2 and FANCI. When the proteins have ubiquitin attached, they can form stable attachments to DNA. Without ubiquitin, however, the proteins only attach to DNA for short periods of time. Using electron microscopy, Tan et al. discovered that large numbers of the modified proteins become tightly attached to damaged DNA, helping to protect it and triggering DNA repair processes.

Understanding the role of FANCD2 in Fanconi anemia could lead to new treatments. FANCD2 and FANCI have similar roles in other cells too. Stopping them from protecting damaged DNA in cancer cells could be used to enhance the success of chemotherapies and radiotherapies.

replication forks and R-loops (*Taniguchi et al., 2002*; *Schwab et al., 2015*; *Wienert et al., 2019*) in the nucleus, and protect nascent DNA at these sites from degradation by cellular nucleases (*Schlacher et al., 2012*). The ubiquitinated form of FANCD2, and also its ubiquitinated partner protein FANCI, become resistant to detergent and high-salt extraction from these foci (*Smogorzewska et al., 2007*; *Montes de Oca et al., 2005*), leading to speculation about the existence of a chromatin anchor or altered DNA binding specificity post-monoubiquitination (*Longerich et al., 2014*).

A recent electron microscopy study revealed a DNA interacting domain that is required for FANCI:FANCD2 binding to DNA (*Liang et al., 2016*). The crystal structure of the non-ubiquitinated FANCI:FANCD2 shows that the monoubiquitination sites of FANCI:FANCD2 are buried and therefore inaccessible in the dimer interface of the complex (*Joo et al., 2011*), suggesting that DNA binding might be required to expose the ubiquitin binding sites. Based on biochemical analyses, non-ubiquitinated FANCI and FANCD2 preferentially bind to branched DNA molecules which mimic DNA replication and repair intermediates (*Longerich et al., 2014*; *Longerich et al., 2009*; *Niraj et al., 2017*); however, how that activates monoubiquitination of FANCI:FANCD2 remains poorly understood. DNA is a cofactor for maximal ubiquitination (*Longerich et al., 2014*; *van Twest et al., 2017*), as is phosphorylation by the ATR kinase (*Tan et al., 2020b*; *Ishiai et al., 2008*).

Here, we have reconstituted the FA pathway using recombinant FA core complex and fluorescently labeled DNA oligomer substrates. We show that once monoubiquitinated, FANCI:FANCD2 forms a tight interaction with double-stranded containing DNA. We report the successful purification of monoubiquitinated FANCI:FANCD2 complex bound to DNA using an Avi-ubiquitin construct, and show that the monoubiquitination does not promote any new protein:protein interactions with other factors in vitro. Instead, we reveal a new role of monoubiquitinated FANCI:FANCD2 in forming higher order structures and demonstrate how monoubiquitinated FANCI:FANCD2 interacts with DNA to initiate DNA repair. Our work uncovers the molecular function of the pathogenetic defect in most cases of FA.

## Results

### Monoubiquitination does not promote association of FANCI:FANCD2 with a panel of proteins previously hypothesized to bind the ubiquitinated form

Mono-ubiquitinated FANCI:FANCD2 (henceforth $I^{Ub}D2^{Ub}$) is the active form of the complex in repair of DNA damage. Many previous studies have speculated about the existence of DNA repair proteins that specifically associate with $I^{Ub}D2^{Ub}$. A summary of these proteins is presented in *Table 1*. Using

**Table 1.** List of proteins containing ubiquitin binding domain that are described or predicted to bind to ubiquitinated FANCD2.

| Protein | Function | Domain | Reference |
|---|---|---|---|
| FAN1 | Nuclease | UBZ4 | (*Kratz et al., 2010*) |
| SLX4 | Nuclease | UBZ1 | (*Lachaud et al., 2014*) |
| FAAP20 | FANCA partner | UBZ | (*Hein et al., 2015*) |
| RAP80 | BRCA1 partner | UIM | (*Castillo et al., 2014*) |
| SMARCAD1 | Chromatin remodeler | CUE | (*Densham et al., 2016*) |
| FANCJ | Helicase | - | (*Raghunandan et al., 2015*) |
| PSMD4 | Protease | UIM | (*Jacquemont and Taniguchi, 2007*) |
| SF3B1 | RNA binding protein | UBZ | (*Moriel-Carretero et al., 2017*) |
| TRIM25 | E3 ligase | RING finger | (*Lossaint et al., 2013*) |
| MCM5 | CMG component | - | (*Lossaint et al., 2013*) |
| BRE | BRCA1 partner | - | (*Wang, 2007*) |
| BRCC | BRCA1 partner | - | (*Wang, 2007*) |
| SNM1A | Nuclease | UBZ | (*Yang et al., 2010*) |
| CtIP* | Nuclease activator | C2H2 zinc finger | (*Murina et al., 2014*) |
| Rev1* | Translesion polymerase | UBZ3 | (*Moldovan et al., 2010*) |

*Indicates not tested in our experiments, because protein not produced in TnT system.

recombinant ID2 or $I^{Ub}D2^{Ub}$ prepared by Avi-ubiquitin purification method (*Tan et al., 2020a*), we sought to directly compare the binding of this panel of ID2-associated proteins. Each of the partner proteins was expressed using reticulocyte extracts (*Figure 1a*), and the majority bound to the ID2 complex as predicted based on previously identified associations (*Figure 1b*). The strongest binding proteins in terms of fraction of protein recovered were SLX4, FAAP20, SMARCAD, FANCJ, PSMD4, SF3B1, MCM5 and BRE. Although SMARCAD and SF3B1 appeared 'sticky' in control experiments, recovery of these proteins was still enriched by FANCD2:FANCI beads compared to background. Luciferase protein was used as a control $^{35}$S-labeled prey-protein. Surprisingly, our main observation was that none of the proteins showed any increased affinity for $I^{Ub}D2^{Ub}$ over ID2 (*Figure 1d*).

## Monoubiquitination clamps FANCI:FANCD2 on DNA

An alternative explanation for the observed increase in association between ID2 and its associated proteins after DNA damage is that $I^{Ub}D2^{Ub}$ has an increased affinity for DNA, which brings the protein into closer proximity to these partners. The majority of ID2 associated proteins are chromatin localized. In order to explore the stability of $I^{Ub}D2^{Ub}$ on DNA, we performed in vitro monoubiquitination reactions in the presence of IR-dye700 labeled 60 bp double-stranded DNA (dsDNA). As previously characterized (*van Twest et al., 2017*), we observed DNA-dependent appearance of monoubiquitinated forms of FANCD2 and FANCI when using recombinant FA core complex components (*Figure 2a–b*). ID2 monoubiquitination readily lead to DNA mobility shifts using EMSA (electromobility shift assay) even at low concentrations, but this was not observed for the unmodified (apo)-ID2 complex in the absence of the enzymatically active FA core complex, or when monoubiquitination-defective K-to-R mutants of ID2 were used in the reaction (*Figure 2c*, lanes 2–4).

Previously, we and others showed that various different dsDNA-containing structures could robustly stimulate ID2 monoubiquitination (*Longerich et al., 2014*; *van Twest et al., 2017*; *Sareen et al., 2012*), but that single-stranded DNA (ssDNA) does not. To determine if monoubiquitinated ID2 had increased affinity for other dsDNA containing structures, we repeated the monoubiquitination reactions in the presence of different IR-dye700 labeled DNA structures. Interestingly, non-ubiquitinated ID2 also exhibited high affinity toward 3' flap DNA structure (similar to a replication fork stalled on the lagging strand), which has been previously observed (*Liang et al., 2016*). The 3'-Flap structure, and each of the other dsDNA containing structures, led to increased ID2 monoubiquitination and increased retention of an EMSA shifted band (*Figure 3*). Conversely,

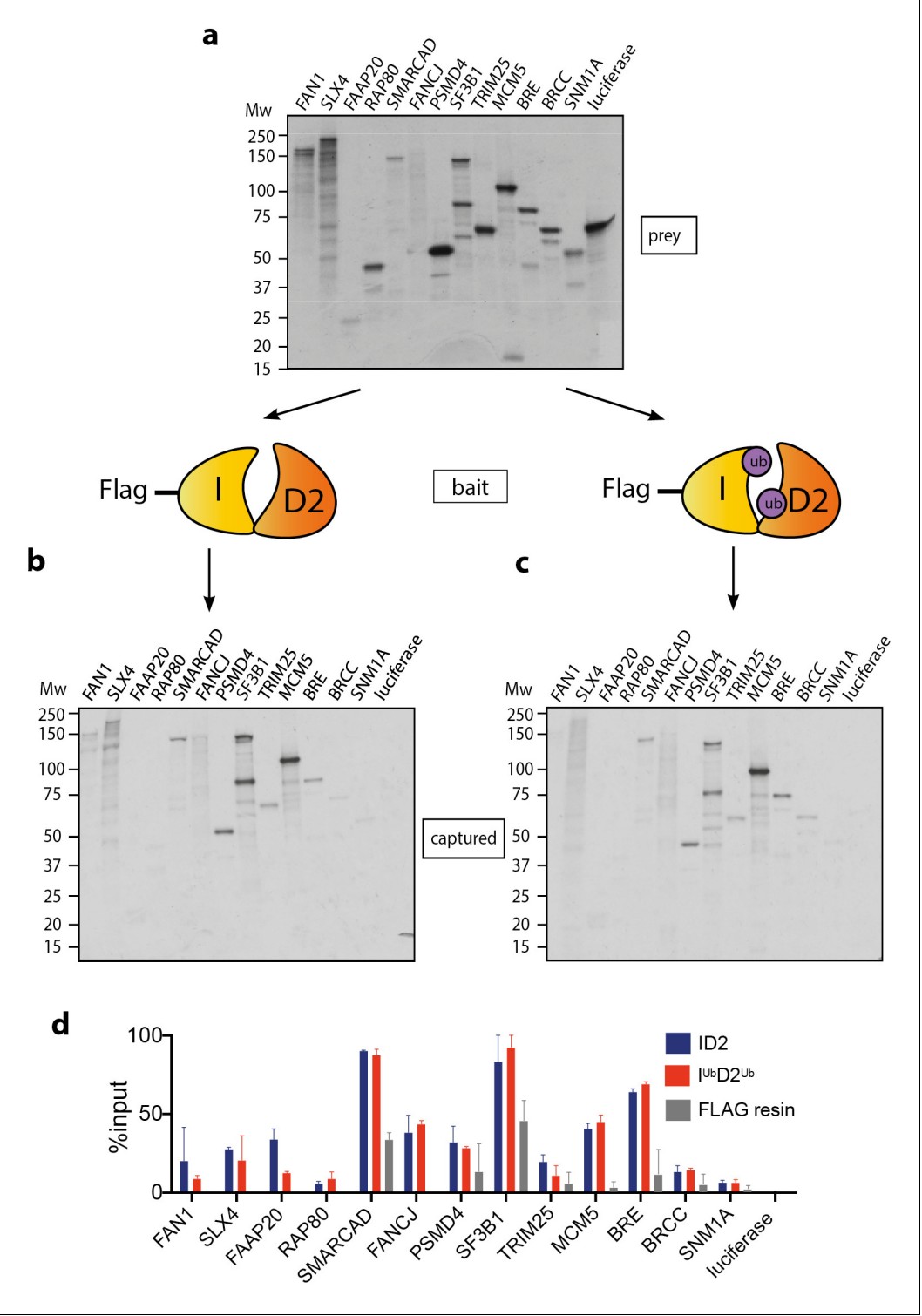

**Figure 1.** Mono-ubiquitination does not alter interaction of FANCI:FANCD2 with DNA repair proteins. (a) $^{35}$S-labelled FAN1, SLX4, FAAP20, RAP80, SMARCAD, FANCJ, PSMD4, SF3B1, TRIM25, MCM5, BRE, BRCC, SNM1A or luciferase (control) inputs were expressed using reticulocyte extracts. (b–c) The inputs prepared from (a) were incubated with the indicated FLAG-ID2 (b) or FLAG-I$^{ub}$D2$^{ub}$ (c) followed by FLAG pull-down and elution. The complexes were subjected to SDS-PAGE, and radiolabelled proteins were detected by autoradiography (representative experiment of n = 2). (d) Quantification showing percentage of ID2, I$^{ub}$D2$^{ub}$ or FLAG resin binding to inputs.

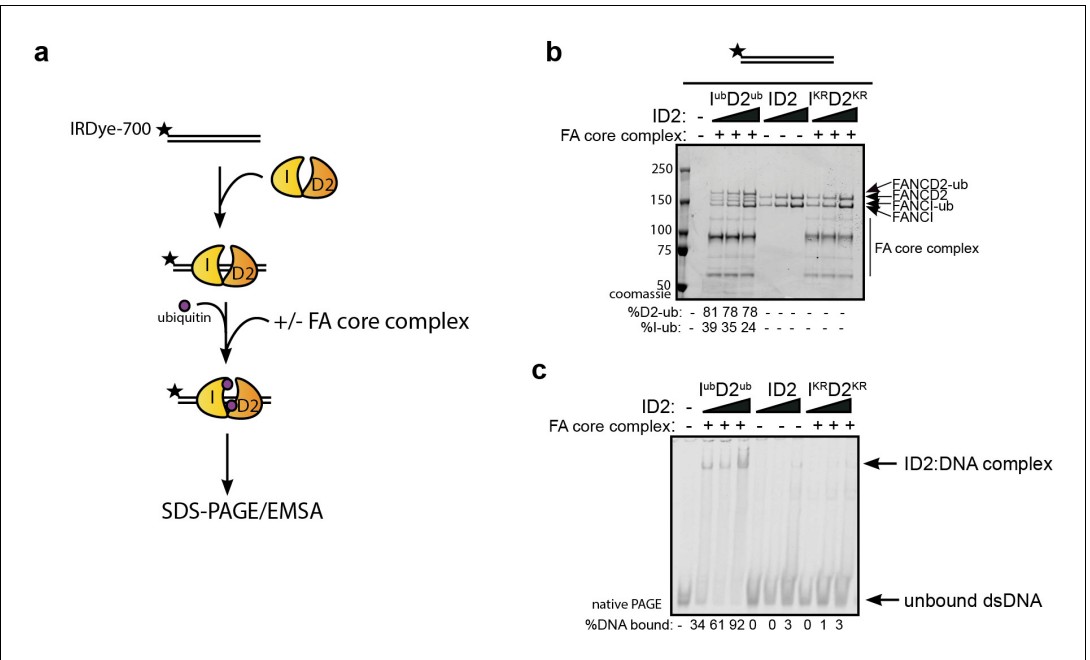

**Figure 2.** Monoubiquitination locks FANCI:FANCD2 on DNA. (a) Schematic of the electrophoretic mobility shift assay (EMSA) using IRDye-700 labeled dsDNA. (b) Coomassie stained SDS-PAGE gel showing monoubiquitination of FANCI:FANCD2 using recombinant FA core complex and IR-dye700 labeled dsDNA. 25, 50 and 100 nM of ID2 or $I^{KR}D2^{KR}$ were incubated with 25 nM of the IR-dye700 dsDNA for 90 min. The respective percentage of FANCI or FANCD2 monoubiquitination were calculated and shown under SDS-PAGE gel. (c) Monoubiquitination reactions from (b) were resolved on 6% native PAGE gel for EMSA analysis. The percentage of ID2 binding to DNA was calculated and shown under native PAGE gel.

ssDNA, which stimulates monoubiquitination more slowly (*van Twest et al., 2017*) but to similar levels at the long time point of this assay, did not cause an EMSA shift.

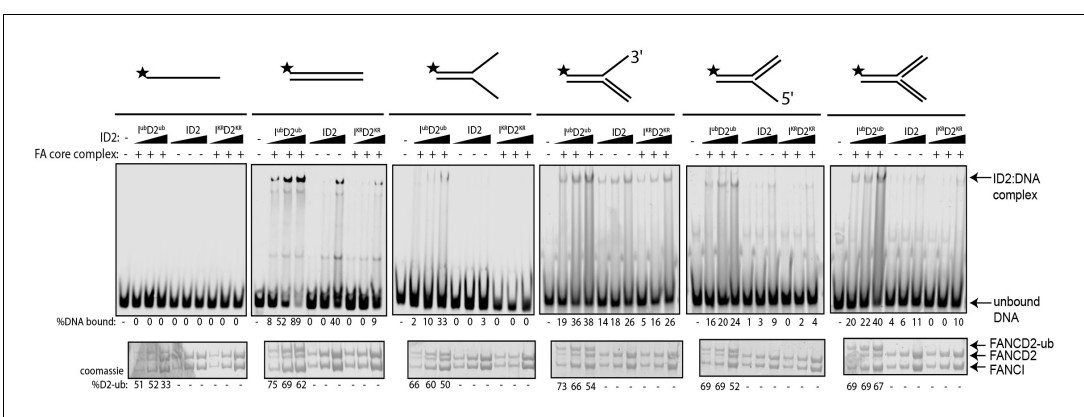

**Figure 3.** Monoubiquitinated FANCI:FANCD2 binds to any type of dsDNA. EMSA gels showing binding of monoubiquitinated or unmodified ID2 complex to different oligo-based DNA substrates. Above each panel, a schematic representing the tested DNA substrate is shown. 25, 50 and 100 nM of ID2 or $I^{KR}D2^{KR}$ were incubated with 25 nM of the indicated DNA substrate and the protein:DNA complexes were resolved on 6% PAGE gels (top). The percentage of DNA binding was calculated and shown under each EMSA gel. Coomassie stained SDS-PAGE gel (bottom) showing the ubiquitination reactions used in the EMSA. The percentage of FANCD2 monoubiquitination was calculated and shown under each SDS-PAGE gel.

# Both FANCI$^{ub}$ and FANCD2$^{ub}$ are associated with a 'clamped' protein: DNA complex

Previous studies reported that monoubiquitination of ID2 complex may lead to dissociation of the heterodimer to its individual subunits, as measured by loss of co-immunoprecipitation of FANCI with FANCD2 (40, 41). In contrast, we did not observe any Ub-mediated dissociation of ID2 in vitro. First, western blotting of the EMSA gels confirmed that the gel shifted DNA band contains both FANCI and FANCD2 proteins (*Figure 4a*). Second, FANCI$^{Ub}$ still co-immunoprecipitated with FANCD2$^{Ub}$ at the plateau of the in vitro ubiquitination reaction (*Figure 4b*).

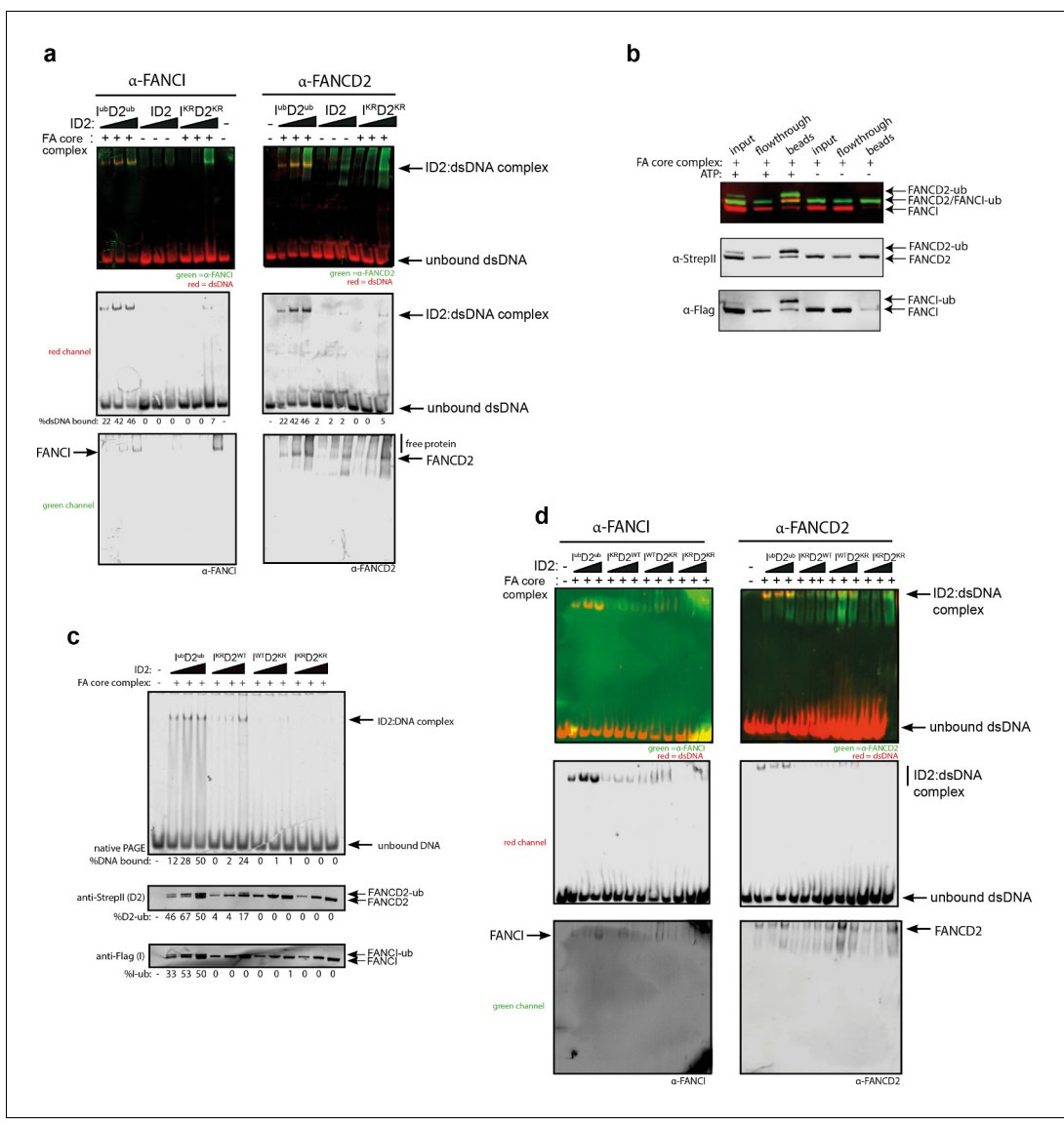

**Figure 4.** FANCD2 monoubiquitination is sufficient for FANCI:FANCD2 locking to DNA, but stimulated by FANCI monoubiquitination. (**a**) Western blots of the EMSA gels containing 50, 100 and 200 nM of I$^{ub}$D2$^{ub}$, ID2 or I$^{KR}$D2$^{KR}$ in the presence of 25 nM IRDye-700 labeled dsDNA (red). Left panels correspond to anti-FANCI antibody (green) and right panels correspond to anti-FANCD2 antibody (green) (**b**) StrepII affinity purification of mono-ubiquitinated (+ATP) and non-ubiquitinated ID2 (-ATP). (**c**) EMSA gels (top) and western blots (bottom) showing the monoubiquitination of 25, 50 and 100 nM I$^{WT}$D2$^{WT}$, I$^{KR}$D2$^{WT}$, I$^{WT}$D2$^{KR}$ or I$^{KR}$D2$^{KR}$ in the presence of 25 nM IRDye-700 labeled dsDNA. (**d**) Western blots of the EMSA gels showing monoubiquitination of 50, 100 and 200 nM I$^{WT}$D2$^{WT}$, I$^{KR}$D2$^{WT}$, I$^{WT}$D2$^{KR}$ or I$^{KR}$D2$^{KR}$ in the presence of 25 nM IRDye-700 labeled dsDNA. FANCI (left, green) and FANCD2 (right, green) remained bound to IRDye-700 labeled DNA (red) after mono-ubiquitination.

To determine the contribution of each of FANCD2$^{Ub}$ and FANCI$^{Ub}$ to the clamping of I$^{Ub}$D2$^{Ub}$ complex to DNA, we used ubiquitination-deficient (KR) mutants in the ubiquitination reaction. FANCI$^{KR}$:FANCD2$^{WT}$ or FANCI$^{WT}$:FANCD2$^{KR}$ mutant results in decrease in EMSA shift, and FANCI$^{KR}$:FANCD2$^{KR}$ did not bind to DNA (*Figure 4c*). However, this retention on DNA correlated with the extent of FANCD2 monoubiquitination retained by these mutant complexes. Western blotting the EMSA gels confirmed that both FANCD2 and FANCI are found in the EMSA shifted product, although in higher amounts when both proteins are capable of being monoubiquitinated (*Figure 4d*).

## Mutant forms of ubiquitin can still clamp ID2 onto DNA

We postulated that the altered affinity for DNA induced by monoubiquitination must result from either a conformational change in the ID2 heterodimer after monoubiquitination, or participation of the conjugated ubiquitin directly in protein:DNA or protein:protein binding. To help distinguish these possibilities, we utilized mutants of ubiquitin that have previously been shown to mediate the known protein:ubiquitin or protein:DNA interactions in other ubiquitinated protein interactions (*Figure 5a*; *Husnjak and Dikic, 2012*). Each of these Ub mutants were conjugated to ID2 by the FA core complex with similar efficiency (*Figure 5b*) and their clamping onto DNA was then measured. Mutations in surface patch 1 (F4A, D58A), surface patch 2 (I44A, V70A), a DNA binding residue (K11R) or a tail mutant (L73P) had no apparent effect on DNA clamping (*Figure 5c*). This result

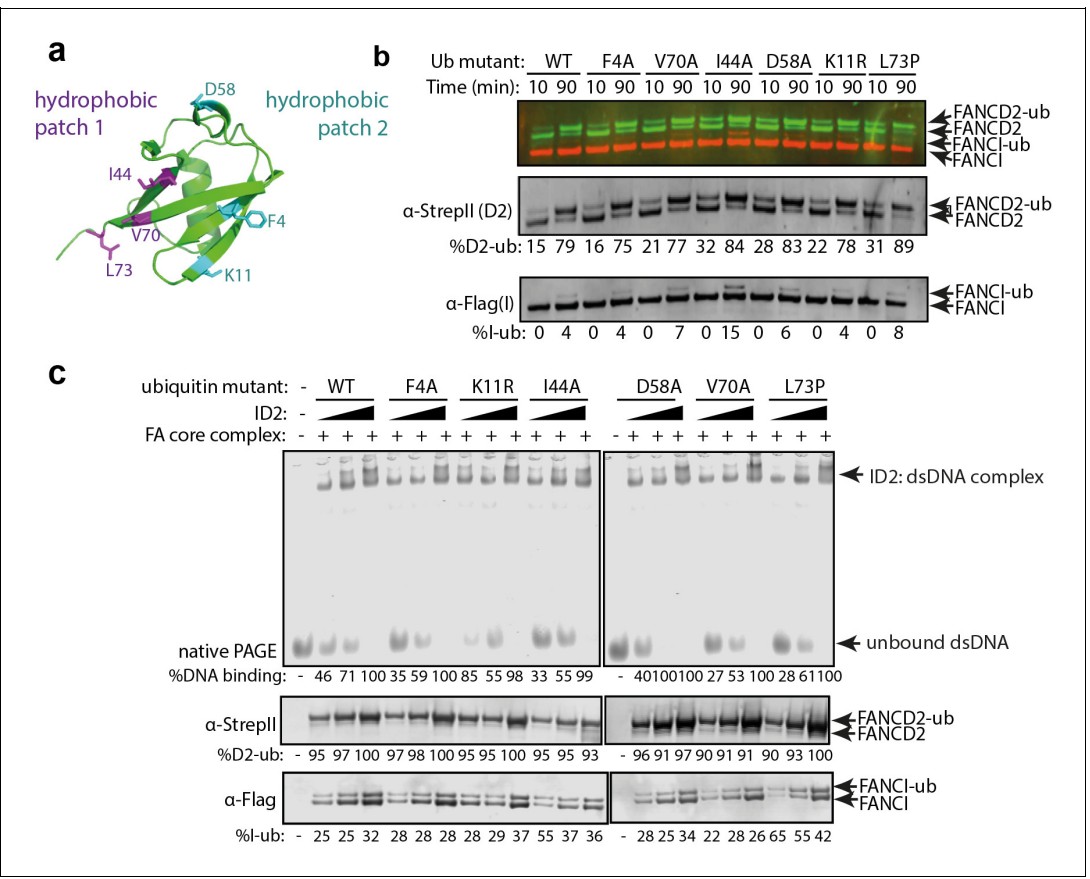

**Figure 5.** Mutations in different ubiquitin patches do not affect ID2 mono-ubiquitination or DNA binding. (a) Crystal structure of ubiquitin with ubiquitin mutant sites depicted (PDB: 1UBQ). Hydrophobic binding pockets are indicated in blue and pink. (b) Western blots showing the time course ubiquitination assays of ID2 using wild-type ubiquitin or ubiquitin F4A, V70A, I44A, D58A, K11R and L73P mutants. (c) EMSA gels showing 25, 50 and 100 nM monoubiquitinated ID2 binding to 25 nM IRDye-700 dsDNA using various ubiquitin mutants (top). Western blots of ID2 ubiquitination products were shown at the bottom and the percentage of FANCI and FANCD2 ubiquitination were shown at the bottom of each western blot panel.

suggests that no canonical surface or region of ubiquitin is critical for DNA clamping of ID2, and instead ubiquitin conjugation to ID2 probably induces a conformational rearrangement of the heterodimer.

## Purification of monoubiquitinated FANCI:FANCD2 complex bound to dsDNA reveals a filamentous architecture

In order to examine the architecture of purified recombinant $I^{Ub}D2^{Ub}$ complex in the presence of dsDNA plasmid, we utilized a recombinant Avi-tag ubiquitin construct containing a 3C protease site between the biotinylated Avi-tag and the N-terminus of ubiquitin (*Tan et al., 2020a*; *Figure 6a*). This tagged ubiquitin is incorporated onto FANCI:FANCD2 by the FA core complex, allowing Avi-din-Sepharose purification of monoubiquitinated ID2 that is then eluted by 3C protease cleavage. We recovered monoubiquitinated FANCI:FANCD2 complex only when FANCI is monoubiquitinated, suggesting that the N-terminus of D2-attached ubiquitin may be buried within the di-ubiquitinated

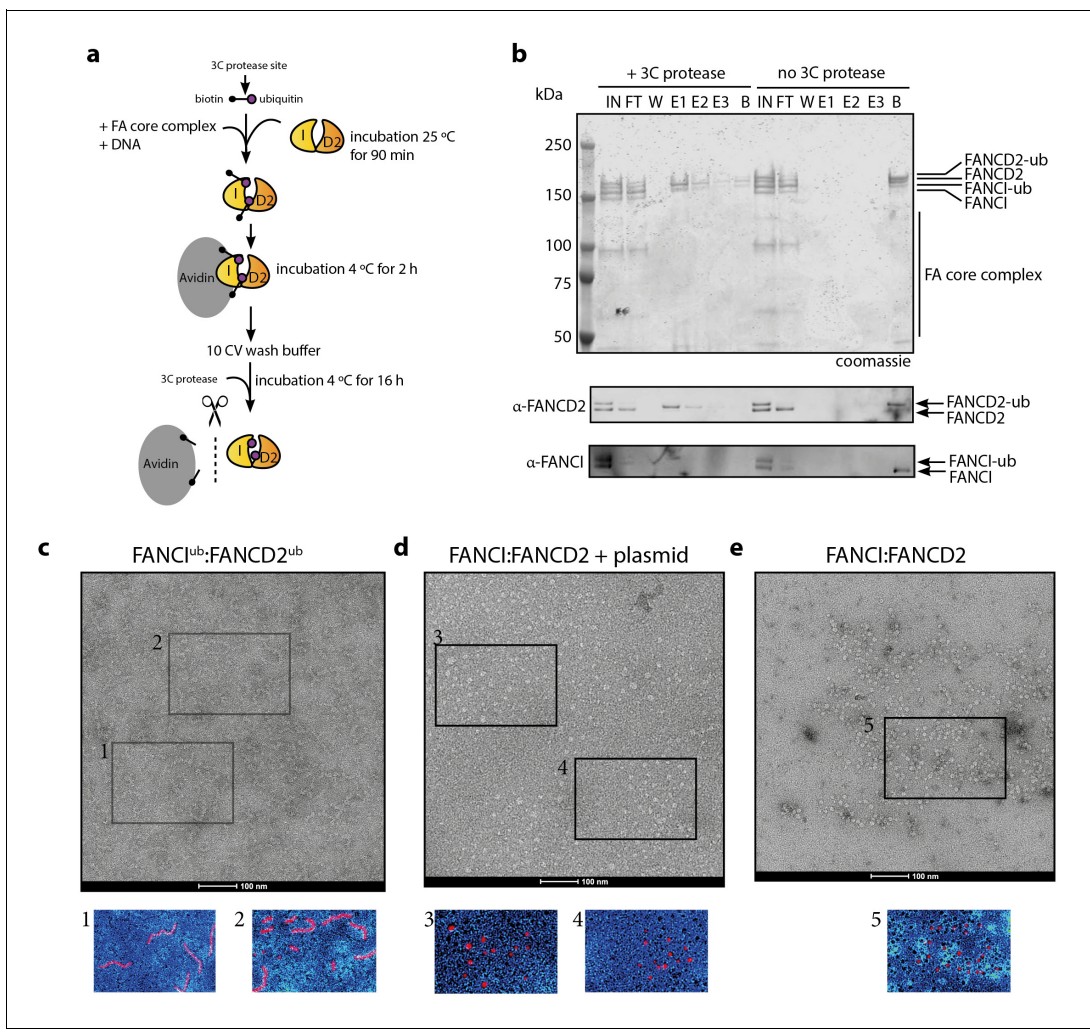

**Figure 6.** Mono-ubiquitinated FANCI:FANCD2 complex assemble into filament-like arrays. (**a**) Schematic of purification of monoubiquitinated FANCI:FANCD2 using Avi-ubiquitin. (**b**) Coomassie stained SDS-PAGE gel (top) and western blots (bottom) showing the purification of monoubiquitinated FANCI:FANCD2 complex eluted using PreScission protease (lanes 1–7) compared to without PreScission protease (lanes 8–14). (**c–e**) Representative negative-stained EM image of purified FANCI^ub:FANCD2^ub complex bound to 2.7 kb plasmid, unmodified FANCI:FANCD2 incubated with 2.7 kb plasmid and unmodified FANCI:FANCD2 complex. Pseudo-colored regions are shown to highlight particular filament-like arrays in FANCI^ub:FANCD2^ub but not other samples.

complex, but the N-terminus of ubiquitin attached to FANCI is accessible for avidin binding (*Figure 6b*).

Using this purified protein, we compared FANCI$^{ub}$:FANCD2$^{ub}$ to unmodified FANCI:FANCD2 using electron microscopy (EM). Surprisingly, we observed that FANCI$^{ub}$:FANCD2$^{ub}$ forms filament-like arrays when bound to dsDNA plasmid (*Figure 6c*). Such arrays were not observed in the unmodified FANCI:FANCD2 protein preparation in the absence of presence of plasmid DNA, nor in previous investigations of human or *Xenopus* FANCI:FANCD2 complexes studied by EM (*Figure 6d–e* and *Swuec et al., 2017*; *Liang et al., 2016*; *Lopez-Martinez et al., 2019*).

When smaller DNA molecules were used as the substrate for ID2 binding, we either observed no filament-like structures (60 bp, *Figure 7a*) or shorter filament-like structures (150 bp, *Figure 7b*) compared to structures that were on average 7-8x longer than the characteristic double saxophone structure of ID2 heterodimer in the non-ubiquitinated state (*Figure 7c*).

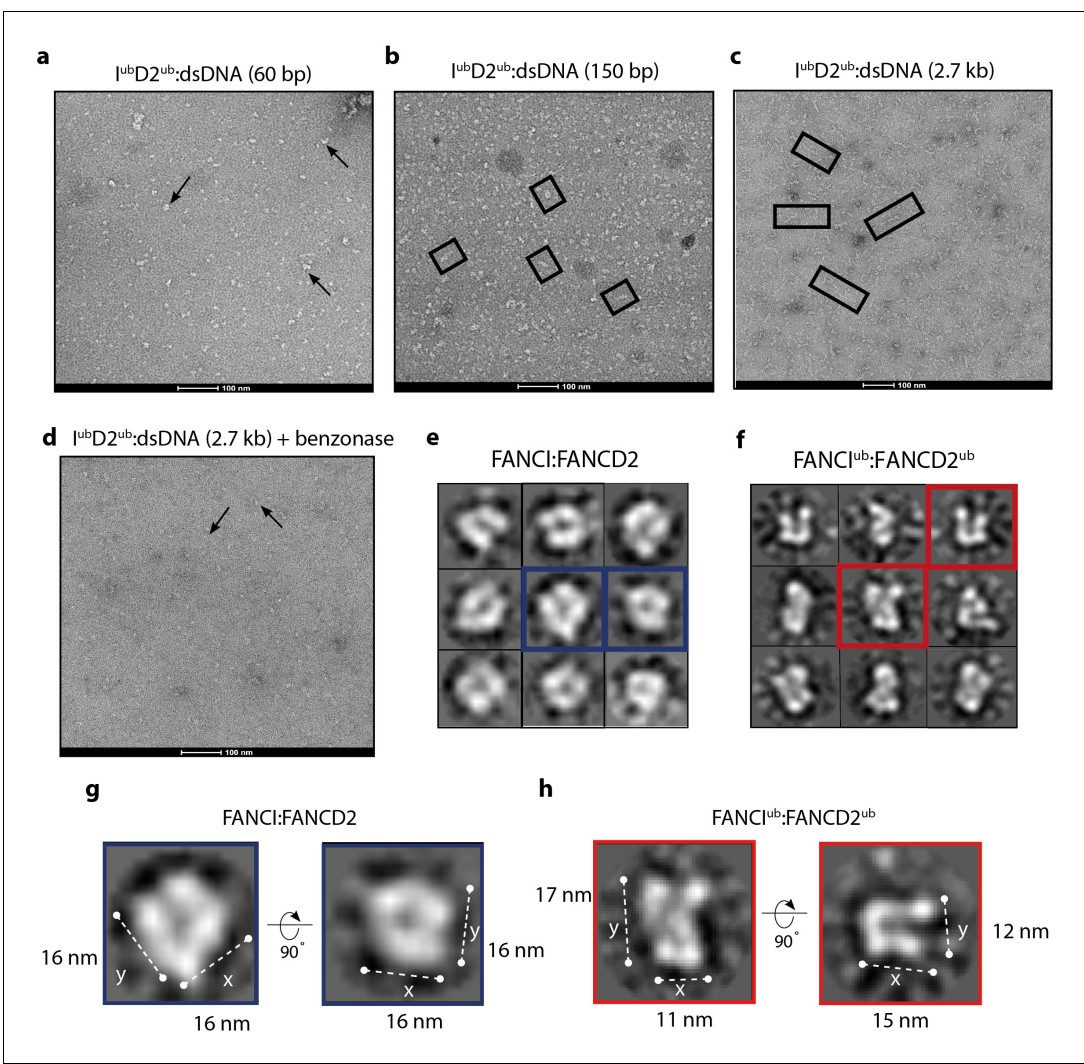

**Figure 7.** Monoubiquitinated FANCI:FANCD2 assembles into filamentous arrays along the length of dsDNA. (a–d) Representative EM image of monoubiquitinated FANCI:FANCD2 bound to (a) 60 bp dsDNA, (b) 150 bp dsDNA, (c) 2.7 kb dsDNA, and (d) 2.7 kb dsDNA and Benzonase-treated. Scale bar, 100 nm. Arrows indicate formation of 1–2 ID2 array; boxes indicate multiple ID2 arrays. (e) Representative 2D class average of ID2. Views of the side and top of ID2 are shown, framed in blue for comparison. (f) Representative 2D class average of I$^{ub}$D2$^{ub}$ bound to 60 bp DNA. Views of the side and top of I$^{ub}$D2$^{ub}$ are shown, framed in red for comparison. (g–h) Example comparison of the length (y) and width (x) of class average images ID2 and I$^{ub}$D2$^{ub}$ (likely an overestimate, because uranyl formate staining increases apparent particle size).

The observation that array length correlated with the size of DNA available for ID2 binding strongly suggested that the association between heterodimer subunits in the array was DNA-mediated. To test whether the array of I$^{ub}$D2$^{ub}$ is also dependent upon binding to the same DNA molecule, we examined the plasmid-stimulated ubiquitination reaction products after treatment with the non-specific endonuclease, Benzonase. It is apparent from EM images that addition of Benzonase breaks the long arrays formed by I$^{ub}$D2$^{ub}$ complex into very short or heterodimer-sized units (*Figure 7d*). This finding is consistent with Benzonase cleaving exposed DNA between I$^{ub}$D2$^{ub}$ units, leading to destabilization of the filamentous arrays. Together our results show that, in vitro, ubiquitination of ID2 leads to a ubiquitin- and DNA- stabilized filament-like structure.

## Single I$^{ub}$D2$^{ub}$ heterodimers on short 60 bp DNA have an altered architecture

Due to variability in the length and shape of filament-like I$^{ub}$D2$^{ub}$ structures on longer DNA molecules we have not been able to uncover the shape or subunit rearrangement of the individual units of the arrays because attempts to class average arrays failed due to a lack of order. However, examination of I$^{ub}$D2$^{ub}$ purified together with short 60 bp DNA allowed us to collect sufficient images of individual particles for analysis. These particles were similar in size to non-ubiquitinated ID2, but it is clear from individual molecule and class average views that the I$^{ub}$D2$^{ub}$ complex forms a distinct architecture from that of ID2 (*Figure 7e–f*). In particular, the overall shape of individual particles and their class averages reveal a twisting that repositions the solenoid arms of one or both of the subunits bringing them into closer proximity. The conformational change induced appears to reduce the size of ID2 in one direction (x vs y) but not the other (*Figure 7g–h*), similar to that predicted in a previously proposed model that placed DNA in a channel between FANCI and FANCD2 post DNA binding (*Longerich et al., 2014*). These images support the view that monoubiquitination induces a conformational change in the ID2 complex that clamps it upon DNA.

## Discussion

The protection of stalled forks by DNA repair factors is essential for proper DNA replication and the maintenance of genome stability. The primary mechanism of replication fork stabilization at interstrand crosslinks, and other replication blocking damage, utilizes the proteins of the FA-BRCA DNA repair pathway. Monoubiquitination of FANCI:FANCD2 by the FA core complex is the central event in this pathway. Here, we showed that monoubiquitination directly clamps the ID2 complex onto double-stranded DNA, and promotes a filament-like coating of long DNA molecules. This finding answers a long-standing question about the nature of the biochemical reaction that is absent in the majority of patients with FA (*Wang and Smogorzewska, 2015*; *Gregory et al., 2003*).

'Fork protection' involves (i) exclusion of cellular nucleases such as MRE11 and DNA2 from the stalled DNA replication fork, and (ii) specific recruitment of other factors that are able to restart DNA replication (*Schlacher et al., 2012*; *Tian et al., 2017*; *Madireddy et al., 2016*). The role of nuclease exclusion seems most important because inhibition of either of two nucleases, MRE11 or DNA2, can significantly alleviate the sensitivity of FA-BRCA mutant cells to replication stalling agents (*Schlacher et al., 2012*; *Tian et al., 2017*). But in many studies, specific association of various repair factors with I$^{Ub}$D2$^{Ub}$ compared to ID2 have been described. These proteins mostly contain ubiquitin-binding domains, providing an impetus for the recruitment-based hypothesis (summary and references in *Table 1*). However, because these previous studies focused on use of ubiquitination-deficient mutants, they could not address the underlying question of whether ubiquitin on either FANCI, FANCD2 or both proteins directly mediated the interaction. Now, we have shown that none of these proteins specifically bind recombinant purified I$^{Ub}$D2$^{Ub}$ compared to ID2. This finding was unexpected, and included proteins such as SLX4 (FANCP), MCM5, and FAN1, which have a demonstrated and essential role in the FA-pathway (*Walden and Deans, 2014*; *Lossaint et al., 2013*; *MacKay et al., 2010*).

Instead, our data provide direct evidence that ID2 undergoes a conformational change after monoubiquitination, that leads to it becoming clamped on double-stranded DNA. It is likely that in previous cell-based experiments specific interaction between SLX4 and FAN1 with FANCD2 but not FANCD2$^{K561R}$ was observed because FANCD2$^{K561R}$ does not become clamped on DNA, even in the presence of active FA core complex. This finding is supported by the observation that FANCD2$^{K561R}$

also does not becomes chromatin localized at damage sites even after extensive DNA damage (*Matsushita et al., 2005*). We propose that I$^{Ub}$D2$^{Ub}$ does not demonstrate restricted interaction with any specific protein partner. None-the-less, its retention in chromatin after ubiquitination would be much more likely to bring the complex into proximity of these other DNA repair factors, where it could still influence their activity.

Clamping onto DNA occurs through a ubiquitin-mediated conformational change in the ID2 complex. A ubiquitin binding-domain (UBD) in FANCD2 has previously been shown necessary for the retention of the protein in the chromatin faction, and for strong binding to FANCI (*Rego et al., 2012*). This UBD domain sits in the FANCD2 structure opposite to where ubiquitin is likely to reside after its conjugation on to FANCI by the FA core complex, and most likely mediates the clamping function and conformational rearrangement. A high-resolution cryo-EM structure of chicken ID2$^{Ub}$ published by Alcón et al alongside our study (*Alcón et al., 2020*), also uncovered a conformational rearrangement of ID2 that is then stapled in place by ubiquitin:UBD association. Other DNA-binding proteins such as histone H2A show an increased association with DNA after monoubiquitination (*Zhang, 2003*) and monoubiquitination also increases the DNA occupancy of transcription factors such as FOXO4 and CIITA (*Greer et al., 2003*). It is possible that ubiquitin to UBD mediated clamping is a general mechanism of protein:DNA target stabilization.

## I$^{Ub}$D2$^{Ub}$ clamped in nucleoprotein arrays

In addition to a conformational change in ID2 induced by monoubiquitination (that has also been concurrently discovered and reported by the Pavletich, Walden and Passmore labs *Alcón et al., 2020*; *Wang et al., 2020*; *Rennie et al., 2020*) we found that monoubiquitinated I$^{Ub}$D2$^{Ub}$ formed large filament-like arrays when it was purified together with plasmid DNA, but not short 60 bp DNA fragments. Fourier Transformation of the EM images did not reveal clear evidence for layer lines, that are expected for fiber diffraction, so we expect that the I$^{Ub}$D2$^{Ub}$ is not a true filament. On average, the length of plasmid-associated structures is 7-8x (but up to 40x) that of that associated with 60 bp DNA. Larger or longer arrays may potentially be obscured from view because the purification strategy makes elution exponentially more difficult with increasing numbers of conjugated ubiquitin-molecules. Steps to remove 'aggregates' may have also inadvertently removed larger arrays. However, as the number of potential plasmid DNA binding sites for ID2 was in large excess the concentration of ID2 used to stimulate reaction, there appears to be some purpose to creation of these filamentous arrays. The modular nature of I$^{Ub}$D2$^{Ub}$ arrays suggests that I$^{Ub}$D2$^{Ub}$ binding to DNA is flexible and can adopt multiple conformations, akin to RPA binding and protecting ssDNA (*Yates et al., 2018*).

There is evidence that I$^{Ub}$D2$^{Ub}$ clamped in nucleoprotein filaments exist in cells. Antibodies against FANCD2 have long been used as a marker of double strand breaks, stalled replication forks and R-loops because the protein forms large, intensely stained foci during S-phase that are increased after treatment with DNA damaging agents (*Taniguchi et al., 2002*; *Schwab et al., 2015*; *Deans and West, 2009*). We suspect that these intense foci are due to coating of DNA around damaged forks, potentially in filamentous arrays similar to those we observed by EM. Support for the large size and extent of DNA binding reflective of filamentous arrays also comes from chromatin immunoprecipitation and sequencing (ChIP-Seq) using anti-FANCD2 (13). FANCD2, and two other damage markers MRE11 and γH2AX, showed no specific localization in a bulk population of cells, but strongly localized adjacent to a Cas9-induced site-specific DNA break. Both γH2AX and FANCD2 produced a broad peak centered at the target site kilobases (kb) to megabases (mb) in length. In contrast, MRE11 is located within a very tight peak within ~100 bp of the break. Chromatin within 1–2 kb of the DSB showed reduced occupancy by γH2AX, consistent with dechromatinization around break sites (*Iacovoni et al., 2010*), but FANCD2 was present right up to the DSB. Accumulation of FANCD2 increases at the DSB early after cleavage, and accumulates more distant from the DSB progressively with time post-cleavage. This is suggestive of a polymerization of the FANCD2 signal away from the break site, as we hypothesize would occur for a protein that forms a growing array of molecules at broken DNA (*Wienert et al., 2019*). The conserved function of FANCD2 as a histone chaperone (*Sato et al., 2012*; *Higgs et al., 2018*) may even be directly linked to displacement of nucleosomes as filamentous arrays extend into break-adjacent chromatin.

In this study, we also observed direct association of two ID2 heterodimers by co-immunopurification only after the protein becomes monoubiquitinated. This approach, if performed in cells, could

be used to further delineate the mechanism and cellular factors required for the extension of $I^{Ub}$-$D2^{Ub}$ arrays during fork protection. Of particular interest will be determining the role of BRCA1 in clamping and/or array extension. BRCA1:BARD1 was initially thought to be the E3 for FANCD2 monoubiquitination, because it co-immunoprecipitates FANCD2, and FANCD2 does not form nuclear foci after damage in BRCA1-deficient cells (*Raghunandan et al., 2015*). However, in various assays it was later shown that FANCD2 monoubiquitination does occur in BRCA1-deficient cells, but it is *uncoupled* from FANCD2 foci formation (*Jacquemont and Taniguchi, 2007*; *Moriel-Carretero et al., 2017*).

## How would a clamped $I^{Ub}D2^{Ub}$ array mediate fork protection?

Filamentous structures on DNA play a genome protective role in prokaryotes: eg DAN protein forms a rigid collaborative filament that reduces accessibility during anoxia (*Lim et al., 2013*), while the *Vibrio cholera* protein ParA2 forms protective filamentous structures on DNA during segregation (*Hui et al., 2010*). Structural characterization has demonstrated how these filaments function and, in the case of ParA2, can be targeted therapeutically (*Misra et al., 2018*). The coating of ssDNA by RPA in eukaryotes, also protect DNA from the activity of nucleases, and directs the specific activity of others (*de Laat et al., 1998*; *Chen et al., 2013*; *Nguyen et al., 2017*). We propose that a FANCI$^{ub}$:D2$^{ub}$ arrays may have a similar stabilizing role on newly synthesized dsDNA at a stalled replication fork. This property would explain why stalled forks are prone to degradation in FA and BRCA patient cells (*Schlacher et al., 2012*; *Tian et al., 2017*). In particular, we hypothesize that filamentous DNA-clamped $I^{ub}$:D2$^{ub}$ could prevent access to DNA by MRE11 and DNA2 nucleases and prevent aberrant ligation of broken DNA to other parts of the genome by non-homologous end-joining.

Second, the tight binding of FANCI$^{ub}$:D2$^{ub}$ to dsDNA, when localized to stalled replication forks, may also prevent the branch migration of replication forks and inhibit their spontaneous or helicase-mediated reversal (*Neelsen and Lopes, 2015*). Reversed forks are the substrate for degradation by DNA2 and WRN nuclease activities, providing a hypothetical link between the activities of FANCD2-monoubiquitination and the nuclease activity of DNA2 and WRN (*Thangavel et al., 2015*; *Sidorova et al., 2013*).

Third, $I^{ub}$:D2$^{ub}$ arrays may also locally suppress non-homologous end-joining (NHEJ) factors, and/or delineate the newly synthesized chromatin from unreplicated regions during the promotion of templated repair processes such as homologous recombination. FANCD2, FANCI, and components of the FA core complex were identified amongst relatively few other factors, in a genome-wide screen for genes that promote templated repair over NHEJ (*Richardson et al., 2017*). Stabilization of RAD51 filaments, required for HR, is also an in vitro property of ID2 (*Sato et al., 2016*), suggesting $I^{ub}$:D2$^{ub}$ filamentous arrays may exist adjacent to or coincident with RAD51 filaments in cells, in order to provide a polarity to the homologous recombination reaction without loss or gain of genomic sequences.

## Role of FANCI-monoubiquitination

Fanci and Fancd2 have common and distinct functions in mouse models of Fanconi anemia (*Dubois et al., 2019*), while the double knockout of FANCI and FANCD2 has an unexpectedly distinct phenotype compared to single knockouts in human cells (*Thompson et al., 2017*). But FANCI$^{K523R}$ expressing cells are less sensitive to DNA damage than FANCI knockout in human cells (*Smogorzewska et al., 2007*), so what is the role of FANCI monoubiquitination? Previous studies demonstrated that FANCI monoubiquitination is always subsequent to FANCD2 monoubiquitination, both in cells (*Sareen et al., 2012*) and in biochemical assays (*van Twest et al., 2017*). FANCI also likely plays a role in recruiting the FA core complex to the substrate (*Castella et al., 2015*). In this study, we show that FANCI-monoubiquitination is not necessary for clamping of the ID2 complex onto DNA (*Figure 4*). However, in vivo it is likely that FANCI monoubiquitination plays a critical role in regulating deubiquitination of the ID2 complex. FANCI recruits the deubiquitinating enzyme USP1:UAF1 (*Yang et al., 2011*), which prevents trapping of monoubiquitinated FANCD2 at non-productive DNA damage sites, but only ID2$^{Ub}$ and not $I^{Ub}D2^{Ub}$ is a substrate (*van Twest et al., 2017*). It is also clear from our EM investigations that FANCI must play an important role in the structural integrity of $I^{Ub}D2^{Ub}$ filamentous arrays on DNA, possibly creating an asymmetry necessary for a specific polarity to array assembly.

## Implications for understanding the deficiency of Fanconi anemia

Onset of progressive bone marrow failure occurs at a median age of 7 in children with FA (*Butturini et al., 1994*). Almost all these patients lack FANCD2 and FANCI monoubiquitination, due to mutation in either *FANCD2* or *FANCI* or one of the nine other FANC proteins required for their monoubiquitination (*Walden and Deans, 2014*). The importance of the monoubiquitin signal is highlighted by the observation that up to 20% of patients acquire somatic reversion of the inherited mutation in a fraction of blood cells (*Soulier et al., 2005*). These mutations restore monoubiquitination and prevent bone marrow failure. Our work suggests two potential strategies for treatment of FA: restoration of gene function, such as that which occurs in somatic revertants or, identification of novel mechanisms to stabilize an ID2:DNA-clamped complex for fork protection by ubiquitin-mediated or innovative means. New small molecule activators or inhibitors of ID2:DNA clamping could be therapeutics in FA or cancer-treatment. In vitro biochemistry has proven to be the most powerful tool in uncovering new functions of FANCD2-monoubiquination that had gone undiscovered for nearly 20 years. The approach is likely to be formidable in drugging the FA pathway in future studies.

# Materials and methods

**Key resources table**

| Reagent type (species) or resource | Designation | Source or reference | Identifiers | Additional information |
|---|---|---|---|---|
| Recombinant DNA reagent (*X. laevis*) | pFastbac1-FLAG-xFANCI | (*Klein Douwel et al., 2014*) | | Gift from Puck Knipscheer |
| Recombinant DNA reagent (*X. laevis*) | pFastbac1-StrepII-xFANCD2 | (*Klein Douwel et al., 2014*) | | Gift from Puck Knipscheer |
| Recombinant DNA reagent (*H. sapiens*) | pFL-EGFP-His-hFANCI | (*Tan et al., 2020a*) | | |
| Recombinant DNA reagent (*H. sapiens*) | pFastbac1-FLAG-hFANCD2$^{opt}$ | (*Tan et al., 2020a*) | RRID:Addgene_ 134904 | Gift from Angelos Constantinou |
| Recombinant DNA reagent (*H. sapiens*) | pFL/pSPL-EGFP-FLAG-B-L-100 | (*van Twest et al., 2017*) | | Codon optimized FANCB |
| Recombinant DNA reagent (*H. sapiens*) | pFL-MBP-C-E-F | (*van Twest et al., 2017*) | | |
| Recombinant DNA reagent (*H. sapiens*) | pGEX-KG-GST-UBE2T | (*van Twest et al., 2017*) | | Codon optimized |
| Recombinant DNA reagent (*E. coli*) | pet16b-Avi-ubiquitin_rbs_BirA | (*Tan et al., 2020a*) | RRID:Addgene_134897 | |
| Recombinant DNA reagent (*E. coli*) | pSRK2706-GST-HRV-3Cprotease | (*Raran-Kurussi and Waugh, 2016*) | RRID:Addgene_78571 | A gift from David Waugh |
| Recombinant DNA reagent (*E. coli*) | pUC19 plasmid | New England BioLabs | N3041S | |
| Strain, strain background (*E. coli*) | BL21 (DE3) | Agilent Technologies | 200131 | |
| Cell line (*Spodoptera frugiperda*) | *Sf9* | Thermo Fisher Scientific | RRID:CVCL_0549 | Maintained in Sf-900 II SFM |

*Continued on next page*

*Continued*

| Reagent type (species) or resource | Designation | Source or reference | Identifiers | Additional information |
|---|---|---|---|---|
| Cell line (*Trichoplusia ni*) | *High Five* | Thermo Fisher Scientific | RRID:CVCL_C190 | Maintained in Sf-900 II SFM |
| Antibody | Rabbit polyclonal antibodies against StrepII | Abcam | RRID:AB_76949 | one in 3000 dilution |
| Antibody | Rabbit polyclonal antibodies against FANCI | Abcam | RRID:AB_74332 | one in 3000 dilution |
| Antibody | Rabbit polyclonal antibodies against FANCD2 | Abcam | RRID:AB_10862535 | one in 3000 dilution |
| Antibody | Mouse monoclonal antibodies against FLAG | Aviva Biosciences | RRID:AB_10884242 | one in 3000 dilution |
| Peptide, recombinant protein | FLAG peptide | Sigma-Aldrich | F3290 | |
| Peptide, recombinant protein | Recombinant Human His6-Ubiquitin E1 Enzyme carrier free | Boston Biochem | E-304–050 | |
| Peptide, recombinant protein | Ubiquitin and associated mutant variants | Boston Biochem | U-110H, UM-I44A, UM-D58A, UM-F4A, UM-L73P, UM-K11R | |
| Commercial assay or kit | TNT T7 Quick Coupled Transcription/ Translation System | Promega Corporation | L1170 | |
| Commercial assay or kit | Anti-FLAG-M2 affinity gel | Sigma Aldrich | RRID:AB_10063035 | |
| Chemical compound, drug | EasyTagL-[35S]-Methionine | PerkinElmer Life Sciences | NEG709A500UC | |
| Software, algorithm | XMIPP | (*de la Rosa-Trevín et al., 2013*) | | |

## Protein purification

Flag-FANCI and StrepII-FANCD2 were expressed using the pFastBac1 vector (Life Technologies). For FANCI:FANCD2 complex, Hi5 cell pellets were resuspended in lysis buffer (50 mM Tris-HCl pH 8.0, 0.1 M NaCl, 1 mM EDTA, 10% glycerol and 1X mammalian protease inhibitor), and sonicated. Lysates were clarified by centrifugation at 20,000 g and the supernatants were incubated with M2 anti-FLAG agarose resin for 2 hr. The resin was washed 5 × 5 min incubation with wash buffer (20 mM Tris-HCl pH 8.0, 0.1 M NaCl, 10% glycerol), and the protein was eluted in the same buffer containing 100 µg/mL FLAG peptide. GST-UBE2T, Flag-BL100, MBP-CEF were purified as described (*van Twest et al., 2017*). Ubiquitin and His-UBE1 were purchased from Boston Biochem.

## Biotinylated-Avi-ubiquitin purification

His-Avi-ubiquitin was purified as described in *Tan et al. (2020a)*.

## In vitro ubiquitination assay

Standard ubiquitination reactions contained 10 µM recombinant human avidin-biotin-ubiquitin, 50 nM human recombinant UBE1, 100 nM UBE2T, 100 nM PUC19 plasmid, 2 mM ATP, 100 nM FANCI: FANCD2 complex wild type (WT) or ubiquitination-deficient (KR), in reaction buffer (50 mM Tris-HCl pH 7.4, 2.5 mM $MgCl_2$, 150 mM NaCl, 0.01% Triton X-100). 20 µL reactions were set up on ice and incubated at 25℃ for 90 min. Reactions were stopped by adding 10 µL NuPage LDS sample buffer and heated at 80℃ for 5 min. Reactions were loaded onto 4–12% SDS PAGE and run using NuPAGE

MOPS buffer and assessed by western blot analysis using Flag (Aviva Biosciences) or StrepII (Abcam) antibody.

## In vitro transcription/translation pull down of $^{35}$S-labeled proteins

Flag-tagged FANCI:FANCD2 and monoubiquitinated FANCI:FANCD2 was prepared by incubating purified FANCI:FANCD2 or monoubiquitinated FANCI:FANCD2 on Flag beads for 2 hr followed by extensive washes in buffer A (20 mM TEA pH 8.0, 150 mM NaCl, 10% glycerol). $^{35}$S-labeled proteins containing UBZ or other ubiquitin domains (Table 1) were generated using the TNT Quick Coupled T7 Transcription/Translation System (Promega) and $^{35}$S-labeled methionine (Perkin Elmer). 10 µL of TNT product was incubated for 4 hr at 4°C in buffer A with 100 ng Flag-tagged FANCI:FANCD2 or monoubiquitinated FANCI:FANCD2, 20 µL of Flag-beads (Sigma-Aldrich) in a 100 µL reaction. Beads were washed five times with buffer A and resuspended in LDS loading buffer. Proteins were separated by SDS-PAGE and visualized by autoradiography.

## Electrophoretic mobility shift assay

Oligonucleotides used to create fluorescently labeled DNA were IRDYE-700-labeled X0m1 (IDTDNA) and other oligos with the sequences shown in Supplementary file 1. Assembly of the different DNA structures was performed exactly as previously described (Supplementary file 1; van Twest et al., 2017). 25 nM DNA substrates were incubated in 20 µL ubiquitination buffer containing 100 nM FANCI:FANCD2, 100 nM BL100, 100 nM CEF, 10 uM HA-ubiquitin (Boston Biochem), 50 nM UBE1 (Boston Biochem) and 100 nM UBE2T at room temperature for 90 min to initiate ubiquitination. The reaction was resolved by electrophoresis through a 6% non-denaturing polyacrylamide gel in TBE (100 mM Tris, 90 mM boric acid, 1 mM EDTA) buffer and visualized by Licor Odyssey system.

## Purification of monoubiquitinated FANCI:FANCD2 complex

Di-monoubiquitinated FANCI:FANCD2 complex was purified as described (Tan et al., 2020a). DNA molecules of 60 bp or 150 bp (dsDNA from oligonucleotides) or 2.6 kb (circular plasmid DNA) were used to stimulate the reaction for different experiments, as indicated.

## Mass spectrometry analysis of monoubiquitinated FANCI:FANCD2 complex

Gels containing monoubiquitinated FANCI and FANCD2 bands were excised and in-gel digested with trypsin and subjected to LC/MS analysis on ESI-FTICR mass spectrometer at Bio21, The University of Melbourne. The analysis program MASCOT was used to identify ubiquitination sites on FANCI and FANCD2.

## Negative stain electron microscopy

Freshly purified monoubiquitinated or non-ubiquitinated FANCI:FANCD2 complex was applied to glow-discharged, carbon/formvar grids and allowed to adsorb for 60 s. Specimen was then stained with 2% uranyl formate for 60 s. Specimen were imaged at a magnification of 73,000 x on a Ceta camera (corresponding to a pixel size of 1.9 Å) in Talos 120 kV. For FANCI:FANCD2 complex, 20 micrographs were analyzed and 4553 particles were picked for 2D classification. For monoubiquitinated FANCI:FANCD2 complex, 20 micrographs were analyzed and 4698 particles were picked for 2D classification. The length and width of 2D class average were measured using ImageJ.

## Single-particle image processing

Monoubiquitinated or non-ubiquitinated FANCI:FANCD2 particles were semi-automatically picked using XMIPP3 (83). The parameters of the contrast transfer function (CTF) for negative stained data was estimated on each micrograph using CTFFIND3 (Mindell and Grigorieff, 2003). Finally, reference free 2D alignment and averaging were executed using XMIPP3.

## Acknowledgements

We are grateful to Alessandro Costa and Paolo Swuec for discussions and suggestions for analysis of the EM data. We thank Puck Knipscheer, Steve West, Johan de Winter, KJ Patel, Paul Hasty, Dario

Alessi, Timothy Richmond, Beverlee Buzon, Andrew Blackford, Steve Jackson and Stephen Elledge for reagents. We thank Eric Hanssen from the Electron Microscopy facility, and Nick Shuai from the Mass Spectrometry facility, at Bio21 Institute, University of Melbourne. WT was supported by an Australian Government Research Training Program postgraduate scholarship. AJD is a Victorian Cancer Agency fellow. WC is an NHMRC career development fellow and Maddie Riewoldt's vision fellow (WC-MRV2016). MWP is an NHMRC Australia Senior Research Fellow. This work was funded by grants from the Fanconi Anemia Research Fund (to AJD and WC), Maddie Riewoldt's Vision (SVI-MRV2017G to WC), the National Health and Medical Research Council (GNT1123100 and GNT1181110 to AJD and GNT1156343 to WC), and the Victorian Government's OIS Program.

## Additional information

### Funding

| Funder | Grant reference number | Author |
|---|---|---|
| Fanconi Anemia Research Fund | | Wayne Crismani<br>Andrew J Deans |
| Maddie Riewoldt's Vision | SVI-MRV2017G | Wayne Crismani |
| Victorian Government | OIS Program | Winnie Tan<br>Sylvie van Twest<br>Rohan Bythell-Douglas<br>Vincent J Murphy<br>Michael Sharp<br>Michael W Parker<br>Wayne Crismani<br>Andrew J Deans |
| Victorian Cancer Agency | | Andrew J Deans |
| National Health and Medical Research Council | GNT1123100 | Andrew J Deans |
| National Health and Medical Research Council | GNT1156343 | Wayne Crismani |
| National Health and Medical Research Council | GNT1117183 | Michael W Parker |
| National Breast Cancer Foundation | IIRS-19-017 | Rohan Bythell-Douglas<br>Andrew J Deans |
| Australian Government Research Training Program | | Winnie Tan |
| National Health and Medical Research Council | GNT1129757 | Wayne Crismani |
| National Health and Medical Research Council | GNT1181110 | Andrew J Deans |
| Maddie Riewoldt's Vision | MRV2016 | Wayne Crismani |

The funders had no role in study design, data collection and interpretation, or the decision to submit the work for publication.

### Author contributions

Winnie Tan, Conceptualization, Formal analysis, Validation, Investigation, Visualization, Methodology, Writing - original draft, Writing - review and editing; Sylvie van Twest, Investigation, Methodology; Andrew Leis, Resources, Software, Supervision, Methodology; Rohan Bythell-Douglas, Formal analysis, Supervision, Investigation, Methodology; Vincent J Murphy, Michael Sharp, Resources, Methodology; Michael W Parker, Conceptualization, Resources, Supervision, Writing - review and editing; Wayne Crismani, Conceptualization, Resources, Formal analysis, Supervision, Funding acquisition, Project administration, Writing - review and editing; Andrew J Deans, Conceptualization, Resources, Formal analysis, Supervision, Funding acquisition, Investigation, Visualization, Methodology, Writing - original draft, Project administration, Writing - review and editing

## Author ORCIDs

Winnie Tan (iD) https://orcid.org/0000-0003-3229-4157
Sylvie van Twest (iD) https://orcid.org/0000-0001-5602-0906
Andrew Leis (iD) https://orcid.org/0000-0003-4905-9401
Rohan Bythell-Douglas (iD) https://orcid.org/0000-0002-3823-8749
Michael Sharp (iD) http://orcid.org/0000-0002-1019-3729
Michael W Parker (iD) http://orcid.org/0000-0002-3101-1138
Wayne Crismani (iD) https://orcid.org/0000-0003-0143-8293
Andrew J Deans (iD) https://orcid.org/0000-0002-5271-4422

## Decision letter and Author response

Decision letter https://doi.org/10.7554/eLife.54128.sa1
Author response https://doi.org/10.7554/eLife.54128.sa2

## Additional files

### Supplementary files

• Supplementary file 1. (**A**) DNA oligonucleotides used in this study. The following oligonucleotides were ordered from IDTDNA. (**B**) Combination of oligonucleotides annealed to generate DNA substrates used in this study.

• Transparent reporting form

### Data availability

All data generated or analysed during this study are included in the manuscript and supporting files.

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
