## [Decision Letter]

**Acceptance summary:**

The authors reconstituted FANCI:FANCD2 ubiquitination by the FA core complex in vitro, and found that monoubiquitination of FANCD2 stabilizes FANCI:FANCD2 heterodimers on dsDNA, and that the purified monoubiquitinated FANCI:FANCD2 forms filamentous arrays in dsDNA. These findings provide some important insights in the role of monoubiquitination of the FANCI:FANCD2 complex in the regulation of the FA pathway.

**Decision letter after peer review:**

Thank you for submitting your article "Monoubiquitination by the Fanconi Anemia core complex locks FANCI:FANCD2 on DNA in filamentous arrays" for consideration by *eLife*. Your article has been reviewed by three peer reviewers, and the evaluation has been overseen by a Reviewing Editor and John Kuriyan as the Senior Editor. The reviewers have opted to remain anonymous.

The reviewers have discussed the reviews with one another and the Reviewing Editor has drafted this decision to help you prepare a revised submission.

Summary:

Thank you for sending your manuscript entitled 'Monoubiquitination by the Fanconi Anemia core complex locks FANCI:FANCD2 on DNA in filamentous arrays' to *eLife*. Your submission was reviewed by three experts in the field, and there is a strong consensus that your study reports significant progress in our understanding of the structure and function of the FANC core complex and has the potential to make a significant impact. However, the reviewers also felt that additional experimentation and analysis is required prior to final acceptance.

Essential revisions:

1) The description of the protocol for FANCI-FANCD2 complex formation requires more information on the exact experimental protocol and detail.

2) More evidence for filament formation by FANCI-FANCD2 is needed to distinguish a filament from other types of oligomeric structures.

For example, biochemical experiments could provide evidence for filament formation. Under conditions of excess DNA, do several protein complexes bind to a single DNA (indicative of a preference of filament formation)? Are there indications of cooperativity?

In addition, it seems that more structural analysis could address this issue. Perhaps 2D averaging of the ID2 filaments could be attempted (since the authors already have made negative stain grids and would probably just need to collect more data). If a repeating unit emerges from the 2D analysis, this would provide a strong argument that the nucleoprotein complex observed is actually composed of a structured filament. This should be tried, but a negative result would not preclude publication.

3) Figure 1 is missing a control of no IDs-complex control for the pull downs.

4) Figure 3: Figure 3, 4th panel from the left: quantification of% DNA bound seems strange. In the gel picture, significant amount of ID2:DNA complex is visible in I/D2 and I-KR/D2-KR lanes, which suggest that this type of DNAs can bind to I:D2 complex even in the absence of ubiquitination. The authors should mention this and show how to interpret this.

5) Figure 4D: Quality of the FANCI blot in this figure is poor, and it is difficult to interpret this data. The result of ID2:dsDNA complex formation shown in Figure 4D is not consistent with the result shown in Figure 4C. (In Figure 4C, less ID2:dsDNA complex is seen in I-WT/D2-KR compared to in I-KR/D2-WT, but in Figure 4D, more ID2:dsDNA complex is seen in I-WT/D2-KR compared to in I-KR/D2-WT.)

6) Figure 8: "reduce the size of ID2 in an X but not Y direction" needs quantification.

---

## [Author Response]

Essential revisions:1) The description of the protocol for FANCI-FANCD2 complex formation requires more information on the exact experimental protocol and detail.

The purification for monoubiquitinated FANCI-FANCD2 and other protein complexes are described in our Avi-Ubiquitin paper just published in PLoS One, Tan et al., 2020. A citation to this paper is now included in the manuscript.

In addition, we have provided a key resource table has been added at the start of the Materials and methods section.

2) More evidence for filament formation by FANCI-FANCD2 is needed to distinguish a filament from other types of oligomeric structures. For example, biochemical experiments could provide evidence for filament formation.

We have attempted several EM and biochemistry approaches but cannot prove that what we see are ordered filaments:

– Fast Fourier Transformation of the EM images does not reveal clear evidence for layer lines, that are expected for fiber diffraction;

– Examination of the power-spectrum relating to boxed-out individual arrays does not reveal evidence of repeating pattern;

– Class averaging of the arrays reveals no evidence of order.

We conclude the I^Ub^D2^Ub^ arrays are probably not true filaments like RAD51/RecA, but more like RPA or SSB, which coat DNA in a cooperative manner but do not form true ordered “filaments”. As such, we have chosen to describe them in this manuscript as “filament-like” or “filamentous arrays”. It will be a major focus of our research group (and others) to follow up on the structure of these arrays and identify mutants or conditions in which array formation is promoted and required. We also think that cryo-EM will be useful to prove whether they are ordered and/or filaments. While obtaining such a dataset is outside the scope of the current manuscript, we have still revealed two novel properties of ubiquitinated-FANCI:FANCD2 complex: DNA clamping (renamed from “locking”), and a higher order structure that coats DNA. As highlighted in the Discussion, these two findings elucidate many unexplained phenomena relating to the Fanconi Anemia pathway and will stimulate a large amount of further research.

Under conditions of excess DNA, do several protein complexes bind to a single DNA (indicative of a preference of filament formation)? Are there indications of cooperativity?

All of the experiments as presented are performed under conditions of excess DNA, so the existence of “arrays” on shorter (150bp) and longer (plasmid) DNA supports the idea of cooperativity. Using our system, it is not a trivial task to perform true cooperativity experiments under properly controlled conditions, because there are so many components. Cooperativity of binding can be measured for something like RAD51 because it is one protein, requiring no rate-limiting modification for DNA binding. But with our system, changing one of DNA, FA core, ID2, and other enzymatic components has indirect effects on all the other components, for example the rate of ubiquitination, that confound the findings of bulk experiments. Instead, we are in the process of establishing a single-molecule system for the observation of individual ID2 particles and their location on a fluorescent DNA molecule prior to and after monoubiquitination. This will allow us, at some time in the future, to elucidate direct observation of cooperativity, but remains a “non-core” part of the findings described in the present manuscript.

In addition, it seems that more structural analysis could address this issue. Perhaps 2D averaging of the ID2 filaments could be attempted (since the authors already have made negative stain grids and would probably just need to collect more data). If a repeating unit emerges from the 2D analysis, this would provide a strong argument that the nucleoprotein complex observed is actually composed of a structured filament. This should be tried, but a negative result would not preclude publication.

As mentioned above, we have attempted for a very long time to 2D averaging the ID2 filaments. The main issue is that the filaments are neither “straight” nor “helical”, thus impeding structural analysis. A very similar problem has been encountered by researchers attempting to structurally characterise filamentous arrays of replication protein A (RPA) on DNA (e.g. see Yates et al., 2018).

3) Figure 1 is missing a control of no IDs-complex control for the pull downs.

We observe no pulldown of any of the proteins tested using only Flag-affinity resin. This data has now been included in Figure 1.

4) Figure 3: Figure 3, 4th panel from the left: quantification of% DNA bound seems strange. In the gel picture, significant amount of ID2:DNA complex is visible in I/D2 and I-KR/D2-KR lanes, which suggest that this type of DNAs can bind to I:D2 complex even in the absence of ubiquitination. The authors should mention this and show how to interpret this.

Thank you for highlighting this observation. We previously observed that 3’Flap DNA was able to stimulate FANCD2 monoubiquitination more strongly than 5’Flap DNA (van Twest et al., 2017), and other studies have also seen increased binding preference for non-ubiquitinated ID2 to 3’Flap (e.g. Joo et al., 2011). We have added more information in the text. We have added this text to the Results: “Interestingly, non-ubiquitinated ID2 also exhibited high affinity towards 3’ flap DNA structure (similar to a replication fork stalled on the lagging strand), which has been previously observed (Liang et al., 2016). The 3’-Flap structure, and each of the other dsDNA containing structures, led to increased ID2 monoubiquitination and increased retention of an EMSA shifted band (Figure 3).”

5) Figure 4D: Quality of the FANCI blot in this figure is poor, and it is difficult to interpret this data. The result of ID2:dsDNA complex formation shown in Figure 4D is not consistent with the result shown in Figure 4C. (In Figure 4C, less ID2:dsDNA complex is seen in I-WT/D2-KR compared to in I-KR/D2-WT, but in Figure 4D, more ID2:dsDNA complex is seen in I-WT/D2-KR compared to in I-KR/D2-WT.)

The FANCI western blot in Figure 4D is now updated with better resolution. The difference in Figure 4D was due to higher protein concentration used for western blot imaging. Figure 4 legend is now updated with the correct protein concentration.

6) Figure 8: "reduce the size of ID2 in an X but not Y direction" needs quantification.

X and Y direction of ID2 and I^ub^D2^ub^ 2D classes are measured and shown in Figure 8C.